# How FaR Are Large Language Models From Agents with Theory-of-Mind?

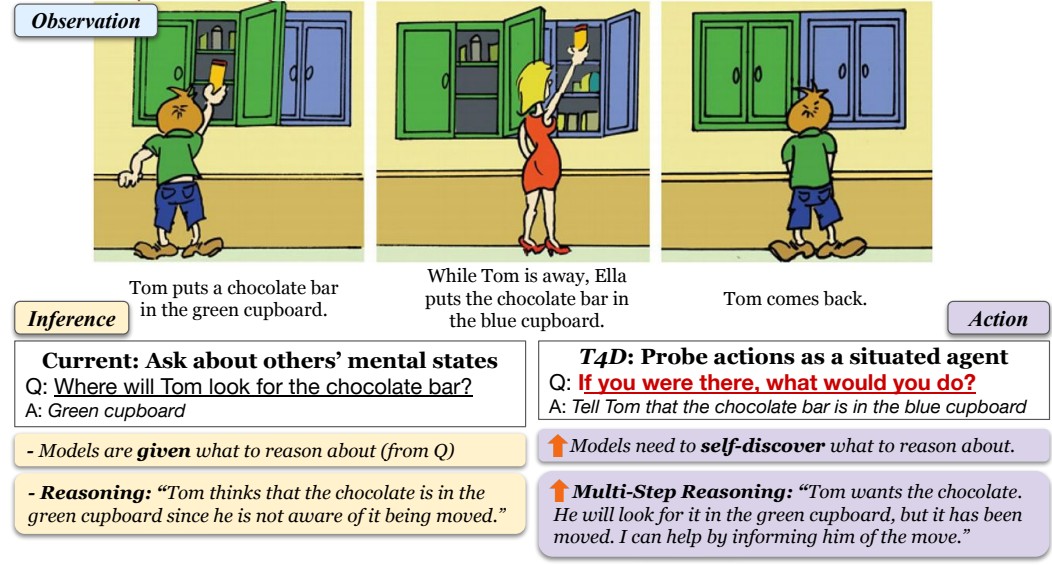

Figure 1: Given *observations*, current social reasoning tasks ask models questions targeting specific *inferences* (left). We propose T4D to probe whether LLMs can decide proper *actions* using theory-of-mind as a situated agent (right). They key challenges in T4D are 1) models have to *identify* relevant inferences about mental states without being directed towards one and 2) to arrive at proper action choices, more steps of reasoning are required.

## Abstract

*Thinking is for Doing.* Humans can *infer* other people's mental states from observations–an ability called Theory-of-Mind (ToM)–and subsequently *act* pragmatically on those inferences. Existing question answering benchmarks such as ToMi ask models questions to make *inferences* about beliefs of characters in a story, but do not test whether models can then use these inferences to guide their *actions*. We propose a new evaluation paradigm for large language models (LLMs): *Thinking for Doing (T4D)*, which requires models to connect inferences about others' mental states to actions in social scenarios. Experiments on T4D demonstrate that LLMs such as GPT-4 and PaLM 2 seemingly excel at tracking characters' beliefs in stories, but they struggle to translate this capability into strategic action.

Our analysis reveals the core challenge for LLMs lies in identifying the implicit inferences about mental states without being explicitly asked about as in ToMi, that lead to choosing the correct action in T4D. To bridge this gap, we introduce a zero-shot prompting framework, *Foresee and Reflect* (FaR), which provides a reasoning *structure* that encourages LLMs to anticipate future challenges and reason about potential actions. FaR boosts GPT-4's performance from 50% to 71% on T4D, outperforming other prompting methods such as Chain-of-Thought and Self-Ask. Moreover, FaR generalizes to diverse out-of-distribution story structures and scenarios that also require ToM inferences to choose an action, consistently outperforming other methods including few-shot in-context learning.

# 1 INTRODUCTION

Humans act with specific intentions, often grounded in reasoning about their environment and the mental states of others. For example, if Tom's friend Anne is looking for her backpack in the office, and Tom knows it is in the kitchen, Tom will intervene to help Anne by suggesting she check the kitchen. This proactive action stems from Tom's understanding of three aspects: 1) Anne's goal of finding her backpack; 2) the knowledge of backpack being in the kitchen; and 3) Anne's belief of thinking the backpack is in the office. Reasoning about Anne's mental states allows Tom to conclude that the mismatch between belief and knowledge prevents Anne from reaching her goal, and his intervention can help. Such capabilities to reason about and act on another individual's beliefs, intentions, and emotions are referred to as Theory-of-Mind (ToM), a critical element of human social interactions (Premack & Woodruff, 1978; Frith & Frith, 2003)

The rise of large language models (LLMs) has prompted extensive research into their potential for *Theory-of-Mind (ToM)* capabilities (Sap et al., 2022; Kosinski, 2023; Ullman, 2023; Shapira et al., 2023a). These investigations predominantly rely on established psychological tests, such as the False Belief Test (Wimmer & Perner, 1983; Baron-Cohen et al., 1985; Perner et al., 1987). While existing benchmarks (Nematzadeh et al., 2018; Le et al., 2019) gauge LLMs' proficiency in *inferring* mental states from scenarios (see Figure 1 left), they often overlook an essential human capability: *acting*[1] *on inferred mental states*. Simply put: humans often act based on inferred intentions and beliefs. In contrast, despite LLMs' performance in the False Belief Test, they often fail to infer what actions would be most useful in scenarios that humans would find trivial, a crucial consideration for the development of next-generation AI agents, from virtual assistants to embodied robots.

We introduce a new evaluation paradigm: *Thinking for Doing (T4D)* (see Fiske, 1992) to probe whether models can *determine proper actions* based on the mental states of others, rather than merely being able to answer questions about mental states. At its core, T4D envisions models as agents processing a series of observations to determine the most apt action from a set of options. Specifically, we adopt stories from a widely-used ToM benchmark: ToMi (Le et al., 2019), based on Sally-Anne False Belief Test (Baron-Cohen et al., 1985) into observations in T4D. This integration ensures that models must utilize mental state reasoning, particularly when a character is identified to hold a false belief (as depicted in Figure 1). The crux of T4D's novelty, as visualized in Figure 1, lies in its objective: instead of merely eliciting inferences from mental state reasoning, it compels models to determine *actions* based on the former.

T4D presents a new zero-shot challenge for LLMs. We find the highest performance (GPT-4) capped at 50% while human annotators reach over 95% agreement. To gain deeper insights into the challenges LLMs encounter in T4D, we identify three reasoning patterns from human-written rationales: question decomposition, theory-of-mind inferences, and commonsense assumptions. Then we test LLMs in *oracle* settings, providing models with oracle reasoning steps based on the identified patterns. As demonstrated in Section 4.2, the primary challenge LLMs face in T4D is pinpointing the correct evidence to inform their actions. When we provide models with specific hints about relevant inferences, their performance significantly improves, approaching human levels.

The clear potential of LLMs to perform T4D with proper guidance leads to the question: Can we develop a method that improves LLMs' T4D performance *without* providing oracle hints but instead teaching models to better *structure* their reasoning process? In response, we introduce a new zero-shot prompting framework **F**oresee **a**nd **R**eflect (FaR) that guides model's inferences by providing a reasoning *structure* using future thinking. FaR has two components: Foresee, where it prompts the models to *predict future events based on observations* and Reflect, where models *reason on which action choice better helps the characters with potential challenges*. Comparison with prompting strategies including Chain-of-Thought Wei et al. (2022), Tree-of-Thought (Yao et al., 2023a) (zero-shot), and Self-Ask (Press et al., 2022) shows that FaR improves LLM zero-shot performance by as much as 50% while other methods do not display significant improvement.

To explore FaR's strengths and limitations in more depth, we perform ablation studies aiming to answer two questions: *are both foresight and reflection needed for improving LLMs* and *what happens if we feed models noisy future predictions*? We find that both components are crucial for tackling T4D and that LLMs are sensitive to noisy reasoning steps about the future in FaR, making how to help

---

[1]We use "*acting*" to refer to performing action in social scenarios like providing information to others.

LLMs *recover* from noisy foresight an intriguing future direction. To examine whether FaR overfits on the ToMi-converted T4D task, we also conduct *generalization study* by testing on out-of-distribution story structures and a non-False-Belief ToM task. We find that FaR shows consistent improvement across generalization tests, even outperforming few-shot prompting. Our contributions are as follows:

1. We propose *Thinking for Doing*, a evaluation paradigm to challenge whether models can connect social reasoning to actions.

2. We find LLMs struggle on T4D and our analysis indicates the key bottleneck is identifying implicit inference steps.

3. We design **F**oresee and **R**eflect (FaR), a zero-shot prompting framework that dramatically improves LLMs' performance on T4D. Analysis and generalization studies show that FaR robustness generalize to diverse contexts.

## 2   BACKGROUND AND RELATED WORK

**Theory-of-Mind and Language Models**   Theory-of-mind has been studied extensively in psychology and cognitive science (Premack & Woodruff, 1978; Baron-Cohen et al., 1985; Frith & Frith, 2003), and clinical psychology tests such as False Belief Test (Wimmer & Perner, 1983) (FBT) were developed to test ToM abilities in children. More recently, as neural language models (LM) display impressive performance in many language understanding tasks, more studies aim to answer whether LMs exhibit ToM (Sap et al., 2022; Kosinski, 2023; Ullman, 2023; Shapira et al., 2023a; Sclar et al., 2023; Trott et al., 2023) using False Belief-templated story datasets such as ToM-bAbI (Nematzadeh et al., 2018) and ToMi (Le et al., 2019). Though stories cover limited range of interactions, other sources of ToM tests also face challenges, such as scalability due to costs of human-generated interactions (Bara et al., 2021) and noises in text-game environments (Zhou et al., 2023). This work focuses on False-Belief tests for ToM, the most studied subarea, and revisits the format of such tasks when testing LLMs. Specifically, while probing work shows that LLMs display some degree of ToM but lack robustness (Sap et al., 2022; Shapira et al., 2022), we find that when asked FBT in a more realistic scenario, models fail even on the unperturbed tasks.

**Large Language Models and Agents**   A line of recent work aims to build *language agents* (Andreas, 2022; Mahowald et al., 2023) that can perform "*actions*". Actions range from mimicking human social behavior (Park et al., 2023), completing tasks using websites (Gur et al., 2023), and tool using (Yao et al., 2023b; Schick et al., 2023). Our work distinguishes from them by focusing on actions that require proper mental state modeling of other individuals (ToM), attributing the performance gap between answering inference questions only and choosing actions based on inferences, and designed a zero-shot prompt that improves models' capability that robustly generalizes.

**Prompting Techniques for LLM**   Recent advancements in the area of LLMs have given rise to a plethora of few-shot (Brown et al., 2020) and instruction (Mishra et al., 2021) prompting techniques, including *Chain-of-Thought* prompting (CoT) (Wei et al., 2022), Least-to-most prompting (Zhou et al., 2022), and search-based approaches like *Tree-of-Thought (ToT)* (Yao et al., 2023a), Graph-of-Thought (Besta et al., 2023; Yao et al., 2023c), and RAP (Hao et al., 2023).

However, the primary objective of our work is not to introduce a new prompting technique. Instead, we focus on the benefits of imposing a structured framework on the LLM's reasoning process, particularly in the context of Theory of Mind (ToM) tasks. Specifically, our analysis (Section 4.2) reveals essential elements of reasoning that can help LLM agents act (Foresee (F) and Reflect (R)), and we capture this in our proposed approach FaR. Moreover, any prompting method that supports granular, multi-step reasoning and captures the Foreseeing and Reflecting steps is well-equipped to address the intricacies of ToM tasks.

## 3   THINKING FOR DOING (T4D): TASK AND DATA

Here we formulate the *Thinking for Doing (T4D)* task that requires models to use social reasoning to choose a proper action as a situated agent.

## 3.1 T4D Task

In grounded social scenarios, an agent's perspective can be distilled into four primary variables: 1. *Observations* $\mathcal{O}$ (e.g., *Tom entered the kitchen. Tom wants a chocolate. Ella moves the chocolate.*), 2. *Task* $\mathcal{T}$ (e.g., *Based on the above observations, who needs help?*), 3. *Inferences* $\mathcal{I}$ (e.g., *Tom is unaware of the chocolate's current location.*), and 4. *Action* $\mathcal{A}$ (e.g., *Inform Tom about the chocolate's location.*). For a comprehensive illustration of these variables in context, please refer to Figure 1.

Traditional social reasoning tasks typically challenge models with questions targeting specific inferences. For example, they might pose a question like "*Where will Jackson look for the onion?*" accompanied by a set of candidate answers (Nematzadeh et al., 2018; Sap et al., 2019; Le et al., 2019). This is depicted in the left side of Figure 1. Formally, this kind of task can be represented as estimation of $P(\mathcal{I}|\mathcal{O}, \mathcal{T}_I)$, where $\mathcal{T}_I$ denotes the inference-directed task articulated by the specific question and its associated answer options.

However, in many real-world AI applications, particularly for embodied agents, decisions often revolve around actions rather than explicit inferences. These decisions are influenced by underlying, often *implicit*, inferences. To bridge this gap, we introduce *Thinking for Doing (T4D)*, a task designed to assess a model's ability to determine the appropriate action based solely on observations, without being directed towards a particular inference. Effectively, T4D represents a shift from directly probing for specific inferences ($\mathcal{T}_I$) to eliciting actions ($\mathcal{T}_A$). In the T4D framework, the model's task is not simply to make an inference but to decide on an action based on inferred mental states. This decision-making process involves estimating $P(\mathcal{A}|\mathcal{O}, \mathcal{T}_A)$, where $\mathcal{T}_A$ encapsulates the action-oriented task, such as determining *Who would you prefer to assist the most?* with potential actions $\mathcal{A}$ like *Assist Jackson* or *Assist Noah*. Crucially, in T4D, inferences $\mathcal{I}$ act as a *latent variable*, inferred from the observable $\mathcal{O}$ to subsequently influence the chosen action $\mathcal{A}$, *i.e.* $P(\mathcal{A}|\mathcal{O}, \mathcal{T}_A, \mathcal{I})$.

## 3.2 Converting ToM Benchmarks to T4D

This study focuses on a critical ability in social intelligence–Theory of Mind (ToM) and converts a widely-used existing benchmark: ToMi (Le et al., 2019) from probing inferences to probing agent's action decisions. In the classic Sally-Anne Test setup (used by ToMi), participants interpret a stroy. For instance, consider Owen who mistakenly believes the suit is placed in the cupboard (Figure 2). ToMi asks models to deduce Owen's mental states, with the expected answer being that Owen will search for the suit inside the cupboard (due to mistaken beliefs).

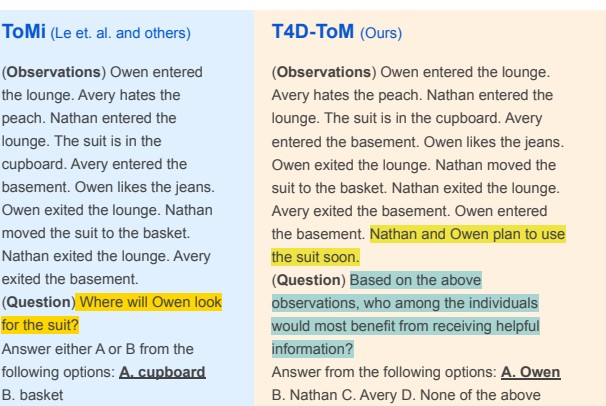

Figure 2: **Task input comparison** of ToMi that asks an inference question given observations and our converted T4D that requires models to choose an action

To shift the focus towards actions as an agent who could potentially intervene and help other characters, we introduce an intent: *both Owen and Nathan intend to use the suit in the near future*. By explicitly stating both characters' intentions, we aim to deter models from adopting a rudimentary heuristic, like automatically assisting the character with immediate plans. However, we also ensure that this complexity does not obfuscate the task for humans. As validated in section 3.3, despite the shared intent to use the suit, human consensus consistently identifies Owen as the one needing help due to his misunderstanding about the suit's location. In our modified task, termed T4D, models are prompted to identify which character they would assist the most by providing accurate information about the onion's location. Thus, in the T4D adaptation, models must deduce from the narrative that: 1) Owen remains under the false impression that the suit is in the cupboard, and 2) considering his impending need for the suit, accurate knowledge about its location would significantly benefit him. We programmatically convert the stories of ToMi (around 500) to T4D due to ToMi's templatic nature. Details of conversion are in Appendix B.

### 3.3 Human Agreement on T4D

Before using T4D to evaluate our models, we seek to verify its validity by testing it with human ToM (*e.g.*, would human ToM encourage helping a character who holds outdated beliefs?). To do so, we randomly sampled around 80 instances for evaluation by $n = 20$ human raters. To ensure this human study reflects how most people would use ToM in real life, we do *not* pre-train these raters extensively on the ToM tasks and do *not* provide any answers in the task examples. Our findings underscore the robustness of T4D tasks: every instance garnered agreement from at least 17 of the 20 raters. Moreover, *over 90% of the instances achieved agreement levels exceeding 95%* (19 or all 20 raters in consensus). This strong human consensus shows that the design of T4D naturally aligns with human perspectives on decision-making.

## 4 LLMs Struggle on T4D While Humans Find it Easy

Here we test LLMs on our T4D task and compare with their performance on the original ToMi set that we convert from. We use PaLM 2 (Anil et al., 2023) Bison (S) and Unicorn (L) [2], ChatGPT (GPT-3.5) (OpenAI, 2022), and GPT-4 (OpenAI, 2023) accessed between June and August, 2023.

### 4.1 Thinking Is "*Easy*", T4D Is Challenging for LLMs

We focus on zero-shot performance following recent studies (Sap et al., 2022; Shapira et al., 2023a; Sclar et al., 2023) to probe LLM's capabilities to understand and use theory-of-mind. Specifically, we provide answer options and instruct models to output one answer option. The results comparing LLM's performance on ToMi and T4D-ToM are shown in Table 1. We find that both PaLM 2 and GPT models perform close to perfect human scores on ToMi (best model GPT-4 gets 93% vs human 100%) but the performance gap enlarges significantly across all models when tested on T4D-ToM (GPT-4

Table 1: **LLMs' accuracy on T4D compared with ToMi**. We find gap between human performance on T4D is much larger than that on ToMi (*we count humans correct when there is more than 95% agreement).

| Models | ToMi | T4D-ToM |
|---|---|---|
| PaLM 2-S (Bison) | 87 | 16 |
| PaLM 2-L (Unicorn) | 87 | 30 |
| GPT-3.5-turbo (ChatGPT) | 74 | 15 |
| GPT-4 | **93** | **50** |
| Random Guessing | 50 | 26 |
| **Human** | **100** | **90*** |

50% vs human 90%). This discrepancy underscores the challenges posed by T4D for even the strongest contemporary LLMs.

### 4.2 What Makes T4D Challenging for LLMs?

To better understand *why* LLMs find T4D challenging, we conducted a study to understand the reasoning processes that humans use to tackle T4D tasks. By collecting and analyzing human-written rationales, we identified distinct dimensions of reasoning that seem particularly challenging for LLMs. Next, we discuss these challenges and experiments with oracle hints to determine if they can indeed aid the models in overcoming these reasoning hurdles. The major reasoning challenges, along with examples and our proposed oracle hints, are summarized in Table 2 and we include example rationales in Appendix C.

**Question Decomposition (QD)** We find that humans often break down the overarching T4D task into more specific follow-up questions such as "*Who might have an information gap?*" and "*What information I can provide?*". This decomposition bridges the gap between the general question and the provided observations. To emulate this in models, we added oracle hints, spotlighting specific information, derived from the decomposition process. Essentially, we guide the models with *oracle* inference results ($\mathcal{I}_Q$), restructuring the task as *i.e*, $P(A|\mathcal{O}, \mathcal{T}_A, \mathcal{I}_Q)$.

**Theory-of-Mind Inferences (ToM)** The second major reasoning challenge is the core inference tested in the Sally-Anne test – can models correctly infer that Sally will look for the item in the *old* location because she left the room before Anne moved the item? We make the **ToM** reasoning

---

[2]https://blog.google/technology/ai/google-palm-2-ai-large-language-model/

Table 2: **Reasoning-Level breakdown**. Following the example task from Figure 2, we show 3 types of reasoning challenges with example specific reasoning steps and design oracle hints to make each challenge *easier* to analyze what makes LLMs struggle on T4D.

| Reasoning Challenges | Example Reasoning Steps | How to Provide Oracle Hints |
|---|---|---|
| Question Decomposition (QD) | *Who would benefit from info?* 
 *–>Nathan and Owen plan to use the suit* 
 *–>Do they know **the suit's location**?* | Add hint after question: 
 "HINT: this information is about 
 an **item's location**" |
| Theory-of-Mind (ToM) | *Nathan and Owen plan to use the suit soon* 
 *–>They need to know the location* 
 *Owen left before the suit was moved* 
 *–>**Owen thinks the suit is in the cupboard*** | Provide oracle ToM inference: 
 "**Owen will look for the suit in 
 the cupboard**" |
| Common Sense Assumption (CSA) | Nathan moved the suit to the basket 
 –>Though not mentioned, we can 
 **assume that the basket is lounge 
 as Nathan is not said to exit the room** | Make assumptions explicit: 
 "**Cupboard and basket are in lounge**" 
 "**Characters do not leave room 
 unless explicitly stated**" |

challenge easier by providing oracle ToM inferences ($\mathcal{I}_{ToM}$) in the observations: "*Sally will look for the [ITEM] in the [OLD CONTAINER]*". This shifts the task to $P(A|\mathcal{O}, \mathcal{T}_A, \mathcal{I}_{ToM})$.

**Common Sense Assumptions (CSA)** The ambiguity inherent in ToMi, as noted by Sclar et al. (2023), presents another challenge. To solve the task, models must assume that both containers are located in the room, even though this is never mentioned explicitly in the observations. We make these assumptions explicit in the observations, *i.e*, $P(A|\mathcal{O}, \mathcal{T}_A, \mathcal{K}_{CS})$, where we use $\mathcal{K}_{CS}$ to indicate commonsense knowledge not explicitly present in the observation.

**Analysis Results** As illustrated in Figure 3, providing oracle hints yields varying results across the identified reasoning dimensions. Guiding models with hints related to item location (+QD) and incorporating oracle-derived character beliefs (+ToM) significantly enhances task performance. In contrast, merely clarifying assumptions (+CSA) has a limited effect on boosting model accuracy.

We hypothesize that providing QD or ToM inferences helps models by supplying *suggestive evidence*, either in the form of leading questions ($\mathcal{I}_Q$) or relevant ToM inferences ($\mathcal{I}_{ToM}$). These results also suggest that the underlying reason for the low performance of LLMs on T4D is

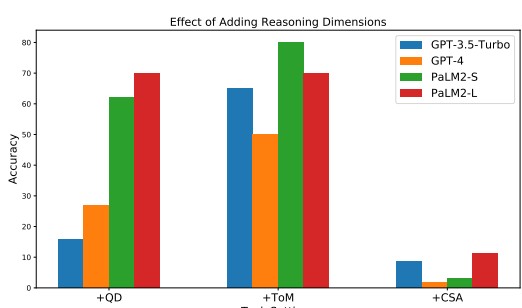

Figure 3: **Increase in performance with provided reasoning levels**. Adding oracle inferences about question decomposition (especially for PaLM2) and ToM dramatically improve performance.

attributed not to the task design but to their failure in drawing correct inferences and reasoning. Thus, a key bottleneck in LLMs that makes T4D challenging (but easy for humans) is navigating the *unconstrained* latent inference space $\mathcal{I}$ to *locate the proper inference* that makes choosing which action intent clear.

## 5 Foresee and Reflect (FaR) Prompting

Building on the insights from our T4D-ToM task analysis, we investigate can we help LLMs *identify an implicit inference path* that leads to correct action choices without hints. Given observations, humans find it natural to identify relevant inferences and arrive at decisions such as "*who should I provide information to?*" However, ensuring that LLMs perform similarly structured reasoning is challenging. Although evidence points to LLMs' ability to infer, they do not necessarily connect these inferences to coherent reasoning about actions.

Our main methodology is to provide LLMs with a generalizable *reasoning structure* that guides the models to relevant inferences. To this end, we introduce the *Foresee and Reflect* (FaR) framework. This framework equips LLMs with a structured reasoning paradigm, prompting them to: 1) extrapolate potential future events from given observations, and 2) introspect on actionable steps that would best

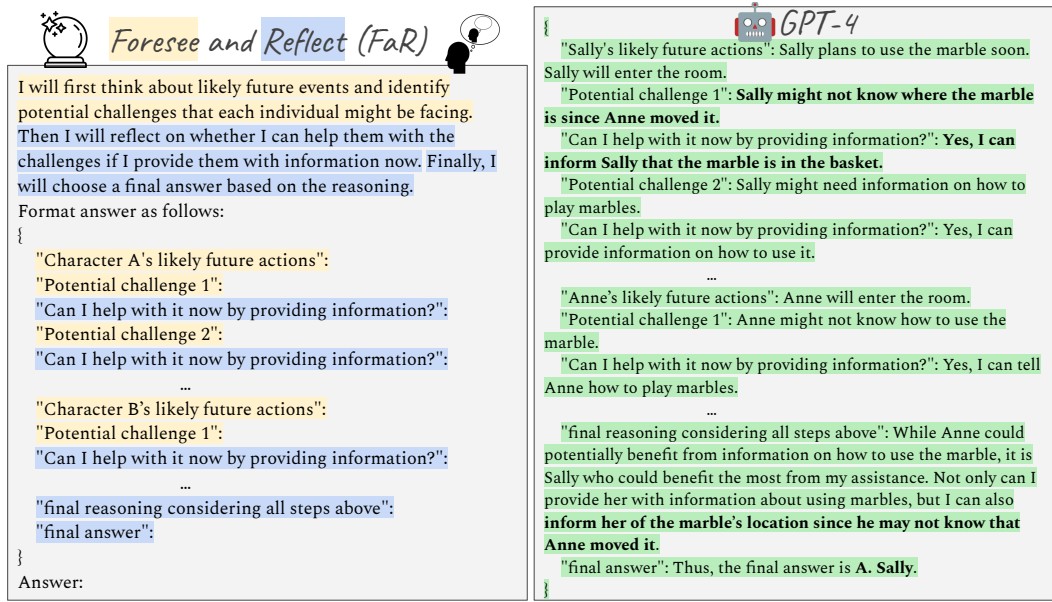

Figure 4: **Foresee and Reflect (FAR)** prompt (left), a new zero-shot prompting framework that combines *future prediction* and pruning by *action-aware reflection*. The *Foresee* part is highlighted in yellow, *Reflect* is highlighted in blue. Example GPT-4 output shown on the right. The model follows FaR and structures intermediate reasoning steps by copying keys and filling in the values so we only need *one* inference call.

serve humans in real-time contexts. As argued in Section 2, the primary contribution of FaR is not to introduce a new prompt but to showcase the benefits of imposing a structured framework on the LLM's reasoning process. Figure 4 presents FaR with an example output from GPT-4.

## 5.1 *Foresee*: CONSIDERING POTENTIAL FUTURE EVENTS

We design FaR by first prompting models to *look into the future* by considering potential events that are likely to happen. This stems from the understanding that the most valuable help often aligns with shaping a more desireable future outcome more desirable. This is also related to a personality trait referred as "*Consideration of Future Consequences* (CFC)" in psychology (Strathman et al., 1994), which is the ability to predict future consequences to inform current action decisions. Given the observations $\mathcal{O}$, FaR guides LLMs to iterate over each character in the narrative, predicting their *likely future actions* and pinpointing the *potential challenges* they might encounter. This approach effectively broadens the initial observations, extrapolating inferences about potential future events.

## 5.2 *Reflect*: REASONING ABOUT ACTIONS

After *foreseeing* likely future events, we prompt models to *reflect* on whether performing actions at the moment could help with the potential challenges identified in the first step. This process can be considered as *pruning* the generated potential future inferences based on the available action options. Overall, FaR helps LLMs connect relevant inferences about future with the intended action choices, completing a reasoning chain spanning *Observation–Inferences–Action*.

**Connection to the A\* Search Algorithm**   The FaR methodology is conceptually analogous to the A\* search algorithm (Hart et al., 1968), an algorithm for finding the optimal path in a weighted graph. We draw the following connections: **Start and Goal**: FaR begins with observations and aims to arrive at an optimal action decision. **Expanding Nodes**: In the *Foresee* phase of FaR, potential inferences (akin to nodes in A\*) are expanded by considering future events. **Heuristics**: The predictions made during the *Foresee* step act as heuristics, guiding the reasoning process toward the most relevant inferences. **Path Pruning**: The *Reflect* stage in FaR narrows down the inferred events based on available actions, similar to how A\* prunes paths based on the heuristic and cost so far.

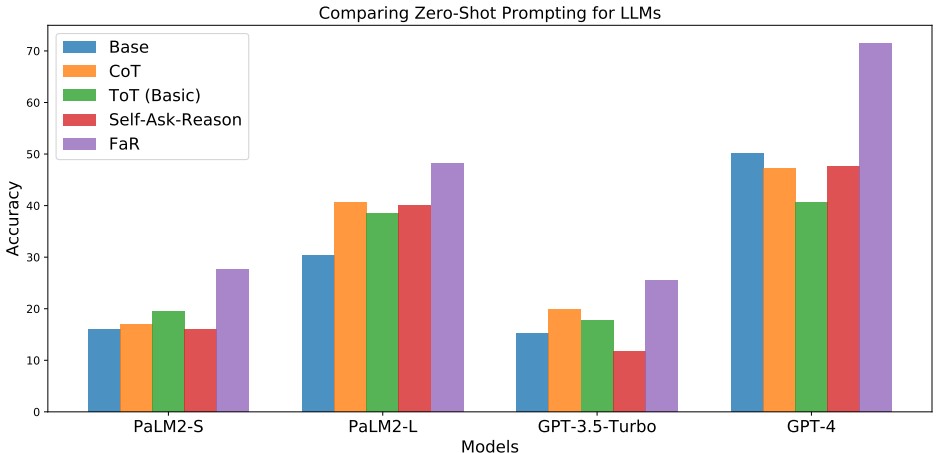

Figure 5: **Comparison of zero-shot prompts on multiple LLMs.** We find FaR improves LLMs performance the most, especially on GPT-4.

## 6    FaR Boosts LLM Dramatically and Generalizes Robustly

We examine the potential of various zero-shot prompting methods on improving LLM's performance on T4D and conduct generalization tests. We aim to answer three research questions through our experiments: 1) *How much can FaR improve LLM's zero-shot performance on T4D?* 2) *Are both the "foresee" and "reflect" components necessary, and what are the limitations of FaR?* and 3) *Does FaR generalize robustly across scenarios where models need to connect inferences with intents?*

### 6.1    Baselines

We consider the following zero-shot prompting strategies, each offering a unique reasoning structure. Full descriptions of the prompts are available in the Appendix D **Chain-of-Thought (CoT)** (Wei et al., 2022):the zero-shot variant from Kojima et al. (2022) and add "*Answer this question by reasoning step-by-step.*" **Tree-of-Thought (ToT)** (Yao et al., 2023a) (Basic Zero-Shot): a zero-shot variant inspired by ToT, which prompts the LLM to envision a discussion among experts. Each expert contributes a reasoning step, and if any expert detects an error in their logic, they exit the discussion. **Self-Ask** (Press et al., 2022): this method emphasizes self-inquiry. Models generate and answer their follow-up questions, iterating until they reach a conclusive answer. A final reasoning step solidifies the conclusion. **FaR**: following Section 5 and Figure 4, we design a prompt that guides models to think about likely future events and challenges that characters might encounter, and reflect whether they can provide help. We apply each prompt and make *one* inference call on all LLMs with maximum 800 tokens with a temperature of 0 (greedy sampling).

### 6.2    FaR Dramatically Improves GPT-4 Zero-Shot Performance

Figure 5 present results of 4 different zero-shot prompting methods. We find that FaR can significantly boost LLMs' performance on T4D-ToM while other prompting methods do not help much. Specifically, FaR helps increase GPT-4 accuracy from base 50% to 71% as well as all other LLMs with the improvement between 12% and 18%. We also observe that more powerful models (GPT-4 and PaLM2-L) tend to benefit more from FaR.

### 6.3    Ablation and Analysis

**Both Foresight and Reflection Are Important**    FaR consists of two main components, one to *foresee* future events and challenges and one to *reflect* on action decisions. To investigate the individual impact of these components, we modified the FaR prompt, isolating each element for ablation. Specifically, we omitted the foresight (referenced as yellow text in Figure 4) and reflection parts (blue text in Figure 4). Table 3 presents ablation on FaR for the two components using GPT-4. We find that the performance significantly drops 17 and 12 points, respectively, when there is no *foresee* and there is no *reflect*, indicating that they are both crucial for T4D.

**Providing Noisy Foresight Undermines Performance**
We further assessed the robustness of the FaR framework by introducing *noisy* foresight. For instance, a spurious foresight for the example in Figure 4 might be "*Sally will enter the bedroom to sleep.*" without any evident reason from the observations. Table 3 shows that LLMs are very sensitive to manually-inputted reasoning steps in FaR and the accuracy of GPT-4 drops from 71% to 42% (even lower than baseline). This highlights a limitation: while the FaR framework can enhance reasoning when guided correctly, it's sensitive to the quality of the foresight provided and can degrade performance if misled.

Table 3: FaR ablations.

| Prompts | GPT-4 Accuracy |
|---|---|
| Base | 50.2 |
| FaR-**NoForesee** | 53.2 |
| FaR-**NoReflect** | 59.7 |
| FaR-**NoisyForesee** | 42 |
| FaR | **71.4** |
| Random Guessing | 26 |
| **Human** | 90 |

## 6.4 FaR Generalizes to Diverse Scenarios

We probe the generalizability of FaR by evaluating its efficacy on *out-of-distribution* scenarios.

**Story Structure Robustness Tests**   We use three challenge sets from Sclar et al. (2023) to test if FaR can generalize to story structures beyond those included ToMi. These sets introduce complexities such as the relocation of two items across two rooms (D1), the involvement of multiple characters with an item (D2), and a single item's movement among four containers (D3) [3]. We convert each set (100 stories each) to T4D-style probes using our ToMi conversion methodology. Table 4 shows results on three types of story-structure change of the ToMi stories. Overall, FaR helps LLMs achieve the highest accuracy compared to other zero-shot prompts on all three generalization tests, for almost all models.

Table 4: Results on story-structure tests. FaR consistently improves the most.

| D1 | | | | |
|---|---|---|---|---|
| Model | CoT | ToT | Self-Ask | FaR |
| GPT-3.5 | **52** | 39 | 26 | **52** |
| GPT-4 | **71** | 29 | 33 | 56 |
| PaLM 2-S | 69 | 85 | 52 | **87** |
| PaLM 2-L | 84 | 92 | 87 | **92** |
| **D2** | | | | |
| Model | CoT | ToT | Self-Ask | FaR |
| GPT-3.5 | 21 | 36 | 44 | **70** |
| GPT-4 | 36 | 34 | 60 | **95** |
| PaLM 2-S | 36 | 39 | 15 | **42** |
| PaLM 2-L | 27 | 15 | 22 | **90** |
| **D3** | | | | |
| Model | CoT | ToT | Self-Ask | FaR |
| GPT-3.5 | 35 | 48 | 9 | **50** |
| GPT-4 | 79 | 76 | 63 | **100** |
| PaLM 2-S | 12 | 20 | 20 | **73** |
| PaLM 2-L | 46 | 37 | 12 | **82** |

**T4D-Faux Pas Case Studies**   To further ascertain FaR's adaptability, we ventured beyond the classic Sally-Anne Test context. We explored Faux Pas scenarios, characterized by individuals inadvertently sharing potentially distressing or unwanted information (Baron-Cohen et al., 1999). We consider Faux Pas, a category of social stories where a person "*says something without considering if it is something that others might not want to hear or know*" (Baron-Cohen et al., 1999), and use 20 expert-curated stories from Shapira et al. (2023b). We convert the original set to T4D by asking models to choose a character from the stories to provide *emotional support* (examples Appendix E). We test GPT-4 with multiple zero-shot prompts as well as few-shot prompting with examples from T4D converted from ToMi. Table 5 shows that FaR outperforms other methods dramatically, showing the generalizability of the zero-shot prompt FaR.

## 7 Conclusion

We propose T4D, a task designed to challenge the capacity of LLMs in bridging Theory of Mind reasoning to actions. Our analyses highlighted a key limitation in LLMs: their difficulty in grappling with implicit inferences without explicit guidance. To mitigate this, we introduced FaR, a structured reasoning paradigm, which not only boosts the performance of LLMs but also ensures broader generalization. As a next step, it would be valuable to delve deeper into understanding the internal representation of LLMs when guided by structured prompts like FaR.

Table 5: Faux Pas results using GPT-4.

| Prompts | Accuracy |
|---|---|
| Base | 31% |
| CoT | 39% |
| ToT | 36% |
| Self-Ask | 43% |
| Few-Shot | 41% |
| FaR | **76%** |

[3]Sclar et al. (2023) propose *SymbolicToM*, a symbolic tracker of mental states for ToMi. We do not include SymbolicToM for comparison in T4D because including answers from the tracker gives away that the model should focus on inferences about *item's location*, whereas other methods are not provided with such assumptions.

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

## A    EXTENDED RELATED WORK

Here we discuss more recent work on probing LMs' ToM capabilities. Gandhi et al. (2023a) proposes BigToM and formalizes probes using causal templates and probes models' forward belief, forward action, and backward belief. T4D differs by probing models action choices by treating them as situated agents and BigToM's action probing is predicting other agents' next actions, which shares similarity to FaR's Foresee step.

Other works have investigated ToM for strategic planning (Ho et al., 2022; Bakhtin et al., 2022; Gandhi et al., 2023b; Dasgupta et al., 2023). T4D differs from this line of work by exposing a key limitation of LLMs even with short-term social reasoning tasks: connecting inference to action is not trivial; we see this insight as orthogonal to ToM for planning and in a strategic environment.

Prompting to analyze models' ToM capabilities have also been recently studied (Moghaddam & Honey, 2023; Trott et al., 2023; Hu & Levy, 2023; Jones et al., 2023). FaR differs by proposing a generalizable reasoning structure focusing on determining agents' own actions instead of direct inference-probing questions.

## B    TOMI CONVERSION DETAILS

ToMi (Le et al., 2019) was proposed as a question answering task based on Sally-Anne Tests and improved upon previous benchmark from Nematzadeh et al. (2018) by removing statistical biases making the task solvable without ToM. Specifically, ToMi defines multiple story primitives such as "*A enters the room*", "*B moves the item*","*A left the room*", etc. and primitives are combined into stories with a finite set of orderings (Sclar et al., 2023). Prior work such as Sap et al. (2022) has found some errors in the ToMi dataset and filtered a clean version that we use to convert to T4D.

On a high-level, conversion consists of two main changes: 1) we add a sentence at the end of the story with the intents of the two characters involved in moving the item ("*Sally and Anne plan to use the marble soon.*"); 2) we propose a new question given the stories about a situated agent's action and provide a list of answer options from all the characters and a "*None of the Above*" option. Specifically, we need to parse the original ToMi tasks to find out 3 types of characters to programmatically generate the additional sentence and new QA options: 1) the character who holds a false belief since they left before another character moves the item. This is also the character who is the correct answer of T4D task, as they benefit the most from receiving helpful information; 2) the character who moves the item after the first character left; 3) distracting characters who do not involve in moving or needing to use the item, as they were introduced in ToMi to reduce biases.

We extract these character names from raw ToMi stories by extracting entities before verbs such as "*moves*", "*enters*", "*leaves*", etc. Then we find the character holding a false belif by extracting from the *original* ToMi questions such as "*Where will Sally look for the marble?*", since the ToMi questions directly targets the character holding a false belief. Next we find who is the character moving the item by extracting the name before "*moves*", and the final character (if any) would be the distracting character. Due to the templatic nature of ToMi, we extract these character names from all stories automatically. Finally, with extracted character names of each type, we apply the changes mentioned above by filling the names.

## C    HUMAN STUDY DETAILS

**T4D Task Setup**    As mentioned in Section 3.3, we conduct human studies with 20 raters who are not trained on T4D tasks and not provided answers in examples. The annotators were selected randomly from a large pool of in-house human annotators. Figure 6 shows a screenshot of our instructions for human study with an example. We provide 3 of such examples without answers and raters answer sampled T4D instances in the multi-choice QA format as well. We also leave it optional for raters to provide their rationales of solving T4D.

Figure 6: Human study instruction screenshot with example.

**Examples of Human Rationales**   In Section 4.2, we summarize 3 reasoning levels from collected human rationales. Table 6 presents examples with highlighted texts corresponding to each of the 3 reasoning levels.

## D   PROMPT DETAILS

Here we show different examples with full zero-shot prompts we use in our experiments. As discussed, we make 1 inference/API call with each zero-shot prompt with maximum tokens of 800.

**Chain-of-Thought**   `You are a clever and helpful AI assistant. Your goal is to help human users around you.`

`One day you have the following observations that follow chronological order:`

`Chloe loves the undershirt. Avery entered the dining room. Aiden entered the dining room. The stockings is in the crate. Avery exited the dining room. Aiden`

Table 6: Examples of human rationales we collected from human study. We highlighted parts from the rationales to correspond to the 3 reasoning levels discussed in 4.2: question decomposition, theory-of-mind inferences (about others' goals and beliefs), and commonsense assumptions.

| | |
|---|---|
| Rationale 1 | Who in the stories have goals that require information? We know that William and Isla both need to use eggplant. They need to know the location of it to be able to use it. William moved the eggplant after Isla exited the closet, thus Isla is not aware of the current location. I should choose Isla. |
| Rationale 2 | William and Isla both plan to use the eggplant. But it is Isla who lacks the knowledge of the current location of the eggplant because William moved it (assuming that both the envelop and the bucket are in the closet. Thus the answer should be Isla. |
| Rationale 3 | Isla is the right answer because she plans to use the eggplant but she does not know that it has been moved to another location. She benefits from me telling her to avoid inconvenience. |

moved the stockings to the cupboard. Aiden exited the dining room. Avery entered the sunroom. Aiden and Avery plan to use the stockings soon.

**Answer the following multiple-choice question by reasoning step-by-step.**

Based on the above information, who among the individuals would most benefit from receiving helpful information? Answer from the following options: A. Chloe B. Avery C. Aiden D. None of the above.

End your answer with 'Thus, the final answer is A (or B or C or D)'

Answer:

**Tree-of-Thought (Basic Zero-Shot)** You are a clever and helpful AI assistant. Your goal is to help human users around you.

One day you have the following observations that follow chronological order:

Sophia entered the front yard. Jayden entered the bathroom. Logan entered the front yard. The cabbage is in the suitcase. Jayden hates the spinach. Jayden exited the bathroom. Logan exited the front yard. Jayden likes the grapefruit. Sophia moved the cabbage to the basket. Sophia exited the front yard. Logan entered the bathroom. Sophia and Logan plan to use the cabbage soon.

**Imagine three different experts are answering this question.**

**All experts will write down 1 step of their thinking,**

**then share it with the group.**

**Then all experts will go on to the next step, etc.**

**If any expert realises they're wrong at any point then they leave.**

**The question is...**

Based on the above information, who among the individuals would most benefit from receiving helpful information? Answer from the following options: A. Sophia B. Jayden C. Logan D. None of the above.

End your answer with 'Thus, the final answer is A (or B or C or D)'

Answer:

**Self-Ask** You are a clever and helpful AI assistant. Your goal is to help human users around you.

One day you have the following observations that follow chronological order:

Lucas entered the cellar. Elizabeth entered the cellar. Ava entered the cellar. The pear is in the basket. Elizabeth exited the cellar. Lucas exited the cellar. Ava moved the pear to the suitcase. Ava exited the cellar. Ava dislikes the slippers. Elizabeth entered the study. Ava and Elizabeth plan to use the pear soon.

Based on the above information, who among the individuals would most benefit from receiving helpful information? Answer from the following options: A. Lucas B. Elizabeth C. Ava D. None of the above.

**I will answer by first coming up and answering useful follow up questions and then reason slowly by considering all the follow up questions and answers, and finally come up with a final answer.**

**Format answer as follows:**

**Are follow up questions needed here: Yes.**

**Follow up:**

**Intermediate answer:**

**Follow up:**

**Intermediate answer:**

**Follow up:**

**Intermediate answer:**

**Let's reason to get a final answer by considering all above follow up questions and answers:**

**So the final answer is:**

End your answer with 'Thus, the final answer is A (or B or C or D)'

Answer:

**FaR** You are a clever and helpful AI assistant. Your goal is to help human users around you.

One day you have the following observations that follow chronological order:

Jacob entered the bathroom. Emma entered the bathroom. The carrot is in the basket. Aiden entered the back yard. Emma exited the bathroom. Jacob moved the carrot to the pantry. Jacob and Emma plan to use the carrot soon.

Based on the above information, who among the individuals would most benefit from receiving helpful information? Answer from the following options: A. Jacob B. Emma C. Aiden D. None of the above.

**I will first think about likely future events and identify potential challenges that each individual might be facing. Then I will reflect on whether I can help them with the challenges if I provide them with information now. Finally, I will choose a final answer based on the reasoning.**

**Format answer as follows:**

**{**

**"Character A's likely future actions":**

**"Potential challenge 1":**

**"Can I help with it now by providing information?":**

**"Potential challenge 2":**

**"Can I help with it now by providing information?":**

**"Potential challenge 3":**

**"Can I help with it now by providing information?":**

**"Character B's likely future actions":**

**"Potential challenge 1":**

**"Can I help with it now by providing information?":**

**"Potential challenge 2":**

**"Can I help with it now by providing information?":**

**"Potential challenge 3":**

**"Can I help with it now by providing information?":**

**"Character C's likely future actions":**

**"Potential challenge 1":**

```
"Can I help with it now by providing information?":
"Potential challenge 2":
"Can I help with it now by providing information?":
"Potential challenge 3":
"Can I help with it now by providing information?":
"final reasoning considering all steps above":
"final answer":
}
```

**Answer:**

End your answer with 'Thus, the final answer is A (or B or C or D)'

Answer:

# E  GENERALIZATION TEST DETAILS

## E.1  STORY STRUCTURE TESTS

Here we show a concrete example for each of the story structure generalization test from Sclar et al. (2023).

**D1: Double Room False Belief Tests:**

Lucas entered the playroom. Alexander entered the playroom. The plum is in the box.

The box is in the playroom. Alexander exited the playroom. Lucas moved the plum to the pantry.

The pantry is in the playroom. Lucas exited the playroom. Alexander entered the porch.

Lucas entered the porch. The strawberry is in the bathtub. The bathtub is in the porch.

Lucas exited the porch. Alexander moved the strawberry to the envelope. The envelope is in the porch.

Alexander exited the porch. Lucas and Alexander plan to use the plum soon.

Based on the above information, who among the individuals would most benefit from receiving helpful information?

Answer from the following options: A. Alexander B. Lucas C. None of the above.

**D2: Three Active Characters Story:**

Lucas entered the playroom. Alexander entered the playroom. Jacob entered the playroom.

The plum is in the box. The box is in the playroom. Alexander exited the playroom.

Lucas moved the plum to the pantry. The pantry is in the playroom. Lucas exited the playroom.

Jacob moved the plum to the bathtub. The bathtub is in the playroom. Jacob exited the playroom.

Lucas, Alexander, and Jacob plan to use the plum soon.

Based on the above information, who among the individuals would most benefit from receiving helpful information?

Answer from the following options: A. Lucas B. Alexander C. Jacob D. Lucas and Alexander E. None of the above.

**D3: Four Containers with Multiple Movements:**

Mia is in the patio. The slippers is in the crate. The crate is in the patio.

Mia moved the slippers to the suitcase. The suitcase is in the patio. Isla entered the patio.

Mia exited the patio. Isla moved the slippers to the bucket. The bucket is in the patio.

Isla moved the slippers to the container. The container is in the patio. Mia and Isla plan to use the slippers soon.

Based on the above information, who among the individuals would most benefit from receiving helpful information?

Answer from the following options: A. Mia B. Isla C. None of the above.

### E.2 FAUX-PAS TESTS

Shapira et al. (2023b) examines LLMs' performance on Faux-Pas stories and release an expert-curated set of Faux-Pas stories. We show an example of the story here: Kim helped her Mum make an apple pie for her uncle when he came to visit. She carried it out of the kitchen.

Kim's uncle is sitting on the couch in the living room. His home has many boxes of pies, most boxes are for pineapple, no apple boxes. He is looking forward to Kim's baking. Kim showed her uncle the pie she made...

Based on the above information, who would you like to help by providing emotional support?

Answer from the following options: A. Kim B. Kim's Mum C. Uncle Tom D. None of the above.

## F   ADDITIONAL RESULTS

### F.1   CONTROLLED EXPERIMENTS WITH TRUE BELIEF

Following several ToM-probing work (Shapira et al., 2023a; Gandhi et al., 2023a), we also test models on a control set where the characters hold true beliefs. Specifically, there is no moving of the item from one container to another and thus they have updated information about the item's location. Notably, since we are probing the model agent's action choices given observations, when all characters hold true beliefs, it is unclear to whom the model agent should help provide information. So, instead of assigning gold labels for the controlled set, we calculate the delta value between the false belief set and the true belief set. The higher delta indicates that the model is more inclined to help when there is a false belief, and the lower indicates that the model might not be using ToM to decide actions space (since it is similarly likely to choose the character with or without false beliefs). As shown in Figure 7, we found that the delta is very low (<5%) for all LLMs when tested 0-shot, and applying FaR helps increase the delta values to around 30%. This indicates that models do not tend to leverage mental state thinking in T4D, but FaR can help them locate relevant ToM inferences to determine the right action options.

### F.2   LLAMA-2 RESULTS

We experiment with a competitive open-sourced LM: Llama 2-70B (Touvron et al., 2023) on T4D and different zero-shot prompting methods. We observe similar drop from ToMi to T4D (63% to 22%). While FaR improves from base, the accuracy is still far from human-level, indicating future direction to improve on smaller open-sourced models.

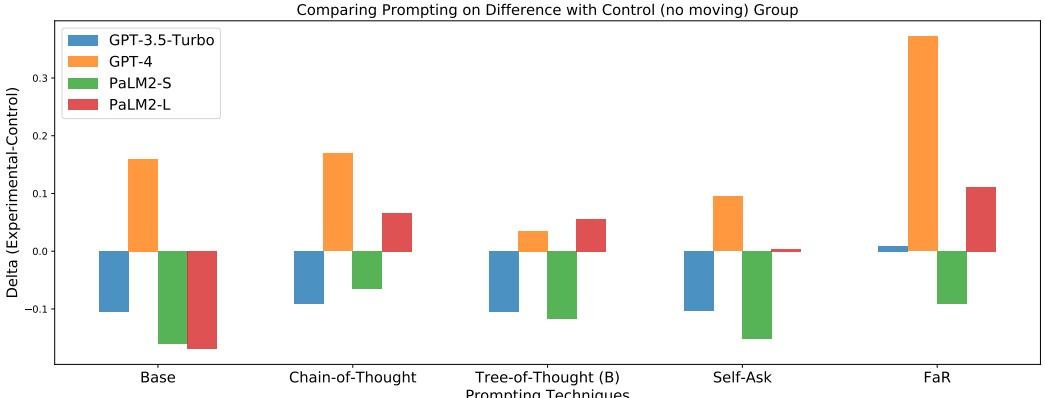

Figure 7: **Delta values of false-belief minus true belief (control set)** Most models receive very low delta values, meaning that their action decisions might not be due to ToM reasoning. After FaR, we observe a dramatic increase in delta, indicating it tends to use ToM reasoning to decide actions on T4D more.

