# OpenReview forum: "How FaR Are Large Language Models From Agents with Theory-of-Mind?"
_ICLR.cc/2024/Conference — Submitted to ICLR 2024_

### Official Review · Reviewer_6i9F · 2023-10-18

**Soundness:** 2 fair
**Presentation:** 3 good
**Contribution:** 3 good
**Rating:** 6
**Confidence:** 4

**Summary:**

This paper proposes a benchmark to test for social reasoning abilities in LLMs. The benchmark they propose is programatically adapted from the existing ToMi benchmark. The novelty of this work over previous work is that the benchmark does not only test for the ability of LLMs to infer unobserved mental states of agents, but tests LLM's abilities to choose an appropriate action for a given scenario based on such an inference. To do this task well, the LLM needs to both do the inference and connect the inference to the appropriate action. The authors show that models can do the ToM inferences well, but struggle on the task of selecting an appropriate action. Humans, by contrast, get near-perfect accuracy.
With further analysis, the paper shows that the problem for models is connecting the inference to the right action; when provided with oracle inferences accuracy goes up significantly. Inspired by this, the work proposes a prompting technique called Foresee and Reflect (FaR). FaR is a zero-shot prompting technique that asks models to reason about the future following from a scenario and use that to reflect on who might need help in the scenario. FaR prompting improves performance for all models tested, but most of all for GPT-4. In further experiments, the authors show that FaR prompting generalises to other datasets, and through ablations they show that both the foreseeing aspect as well as the reflecting aspect are crucial for performance.

**Strengths:**

This paper is well-written and easy to understand.

The authors show convincingly that their prompting method is better than baselines; they use a set of baselines and show it works on multiple datasets.

The analysis by adding information to the prompt to find when models start to perform better on the action selection is insightful.

The ablations they do throughout the work are helpful for understanding the importance of all parts of the prompting method.

The results are interesting; LLMs can do ToM inferences but seemingly cannot connect them to appropriate actions without structured prompting or explicit mentioning of the inferences.

**Weaknesses:**

The authors mention that the main contribution of this work is evaluating the ability of LLMs to take actions based on ToM inferences, but do not mention a paper that does the same: "Understanding Social Reasoning in Language Models with Language Models", Gandhi et al. 2023. This work also tests the ability of LLMs to take appropriate actions in Sally-and-Anne-type scenarios, evaluate humans on it as well, and find similar results (models struggle more with the actions than with the inferences).

The authors apply no control conditions, like true belief situations, to further investigate whether models are actually doing ToM-reasoning (which is common in research investigating ToM in LLMs).

The authors compare models zero-shot to humans who are given three examples of the task. Even if the examples have no label, prior work shows that few-shot prompting without labels often works as well as with labels.

Not necessarily a weakness but I'd like to see some discussion; could there be spurious correlations in the examples? E.g. in Figure 2 Owen is mentioned more often than the others and is also the answer.

To summarise; my main concern with this paper is contribution in light of another work doing the same.

**Questions:**

1. Could you discuss your contribution in light of Gandhi et al. 2023?
2. Why don't you apply control conditions like true belief situations?
3. Why do you give humans 3 examples and models none?
4. Did you test for spurious correlations between the observations and the answer?

Some less important style bits that I'm adding here because they are not weaknesses:
1. The use of italic is confusing and distracting, you sometimes use it for stressing things, other times for spelling out acronyms, other times for no apparent reason
2. Typo on page 4 s/stroy/story
3. Personal preference: figure 3 is confusing and makes it hard to compare models because it depends on initial performance (cannot increase a lot if it was already high); would consider making a different type of plot where models are still comparable, e.g. just highlight the improvement part of a bar in a normal absolute performance plot
4. I don't understand the reason for putting a paragraph about the connection to A*
5. Figure 5 has a confusing title and I would add more details in the caption about what is going on in this figure

---

> ### Author Response · Authors · 2023-11-19
> **Author Response**
>
> Thank you so much for your helpful feedback and suggestions. Our responses to questions are as follows:
>
> __[Main concern with this paper is contribution in light of Gandhi et al. 2023. doing the same]__
>
> Gandhi et al. 2023. proposes BigToM, which is an excellent benchmark for measuring ToM in LMs. While there is a surface-level similarity in the task setups (both contain “action” probing), there is an important distinction between action probing in BigToM and T4D: probing beliefs of others’ actions such as “*What will A do?*” (BigToM) vs. probing model’s own action decisions based on beliefs of others such as “*Who should you provide help with?*” (T4D).  Specifically, in T4D, models have to treat themselves as situated agents and decide their **own** actions, based on potential reasoning about others’ mental states as well as predictions of others' actions. We argue that these are complementary directions, and both are important to understand LLMs’ capabilities related to ToM. In summary, T4D builds on previous benchmarks by focusing on models’ capabilities to determine actions for themselves based on mental state reasoning. We have added this in the extended related work section in the updated draft in Appendix A.
>
> (Gentle note that according to the last question in FAQ from ICLR Reviewer Guide (https://iclr.cc/Conferences/2024/ReviewerGuide), “if a paper was published (i.e., at a peer-reviewed venue) on or after May 28, 2023, authors are not required to compare their own work to that paper.” and “they may be excused for not knowing about papers not published in peer-reviewed conference proceedings or journals, which includes papers exclusively available on arXiv.” and Gandhi et al. 2023. Was released on arXiv on June 21, 2023).
>
> __[Controlled experiment with true beliefs]__
>
> Thanks for the great point! We did controlled experiments with true beliefs (i.e., the item stays in the same location) paired with all our experiments with false beliefs. We have added these results in the appendix F. Notably, since we are probing the model agent's action choices given observations, when all characters hold true beliefs, it is unclear to whom the model agent should help provide information. So, instead of assigning gold labels for the controlled set, we calculate the delta value between the false belief set and the true belief set. The higher delta indicates that the model is more inclined to help when there is a false belief, and the lower indicates that the model might not be using ToM to decide actions space  (since it is similarly likely to choose the character with or without false beliefs). We found that the delta is very low (<5%) for all LLMs when tested 0-shot, and applying FaR helps increase the delta values to around 30%. This indicates that models do not tend to leverage mental state thinking in T4D, but FaR can help them locate relevant ToM inferences to determine the right action options.
>
> __[Human studies include 3 examples while models do not]__
>
> We tried prompting models zero-shot with 3 examples and observed no clear difference from Table 1 (+/-2%). As for humans, we reran human studies with a subset that we do not show any task examples and also found similar results (90% of instances reached 95% agreement on the correct answer).
>
> __[Could there be spurious correlations in the examples?]__
>
> ToMi was extended from an earlier work [1] that uses QA to probe ToM using sally-anne tests by removing biases from the character name mentions (discussed in Section 3 of the ToMi paper [2]). Furthermore, we included an observation-answer-only baseline without the questions and found close-to-random performance, confirming that there is no strong spurious correlation between the observation and the answers.
>
> __[Suggestion on figures and formatting]__
>
> Thank you very much for the constructive suggestions! We will be sure to address them in the next version of the paper.
>
>
> [1] Aida Nematzadeh, Kaylee Burns, Erin Grant, Alison Gopnik, and Tom Griffiths. Evaluating theory of mind in question answering. In Proceedings of the 2018 Conference on Empirical Methods in Natural Language Processing, pp. 2392–2400, 2018.
>
> [2] Le, Matthew, Y-Lan Boureau, and Maximilian Nickel. "Revisiting the evaluation of theory of mind through question answering." Proceedings of the 2019 Conference on Empirical Methods in Natural Language Processing and the 9th International Joint Conference on Natural Language Processing (EMNLP-IJCNLP). 2019.

---

> > ### Comment · Reviewer_6i9F · 2023-11-20
> > **Thanks for addressing my points**
> >
> > Thanks for the detailed rebuttal. Let me start my main weakness, which is the contribution in light of Gandhi et al.
> >
> > I don't buy the distinction between answering questions about ones "own actions" or "other's actions" for an LLM, this seems like a distinction in prompt wording and not in task. However, thanks for pointing out the reviewer guidelines. These make it clear that I should not have given a low score for contribution since the works are concurrent. I'm sorry about that. I'm going to increase my contribution score and with it my rating.
> >
> > The other answers address my concerns, thanks!

---

> > > ### Author Response · Authors · 2023-11-23
> > > **Thanks for considering our response!**
> > >
> > > Thanks again for your feedback, and for taking the time to consider our response. We are pleased to see that we were able to address your concerns. We will add the suggested changes in the next version.

---

### Official Review · Reviewer_WCD3 · 2023-10-31

**Soundness:** 2 fair
**Presentation:** 3 good
**Contribution:** 1 poor
**Rating:** 3
**Confidence:** 5

**Summary:**

The paper proposes a new benchmark, Thinking for Doing (T4D), to test theory of mind capabilities in large language models (LLMs). It modifies the existing ToMi benchmark to require models to not just infer mental states but act based on those inferences. Experiments are conducted with GPT-4 and PaLM-2.  To improve model performance on T4D, the authors introduce a new prompting method called Foresee and Reflect (FaR) which guides the model through two steps: predicting future events and reflecting on actions to address potential challenges.

**Strengths:**

- I found the paper to be very clear, with sections and methods well motivated.
- The paper collects representative data from human participants to measure performance.
- The proposed method improves performance on the given task.

**Weaknesses:**

The paper’s contributions can be split into two parts: A benchmark for testing theory of mind, and a prompting method to achieve. In terms of research, the paper proposes a new problem and a new method. I talk about their weaknesses separately.

### Weakness: T4D

- The formal reasoning steps and inferences required for T4D are not clearly specified. For a model to succeed at T4D, it likely needs to make inferences about agents' utilities, beliefs, perceptual access and how helping might alter beliefs to increase utilities. However, the paper does not delineate the full reasoning process of inferring and comparing utilities across agents that would lead to selecting the correct action. Clearer formulation of the multi-step inference process needed could better highlight the reasoning challenges in T4D.
- Theory of mind relates to having a causal model for other agents’ mental states, how they are formed and how they affect actions (that differ from the self). T4D which is based on ToMi suffers from the same limitations as ToMi:
    - lack of diversity semantic: beliefs are only about locations of objects. Variations in app D. are still very limited.
    - lack of diversity syntactic: all examples have the same structure in the sentences
    - load on memory tracking, rather than testing for Theory of Mind:
    - ToMi does not have clear percepts
    - It is unsurprising that adding these additional “hops“ to reasoning over ToMi and planning leads to lower zero-shot performance.
- Lack of probing methods: the authors only try a single method of probing LLMs; Asking a direct question. The work would improve if alternate probing methods were also tried. For example asking the model to complete a sentence; ie actually do instead of what it thinks it would do.
- The paper ignores work that relates to using theory of mind for planning, ie, in strategic settings [6, 7, 8]; where thinking is for doing. This includes work that relates to collaboration and competition in strategic multi-agent settings. A discussion on this is needed. A paper from human ToM reasoning [5] could be insightful to include in the discussion.
- Similarly, a justification of how this benchmark builds over other Theory of mind benchmarks [ones in the paper and 1, 2, 3] is needed.
- Lack of evaluations: The probing method that they use is restricted to be questions about helping others by providing information.
- Only test with GPT-4 and Palm-2, both closed source models.

### Weakness in methods

- Not a fair comparison across methods. The prompt for FaR is much longer and is much more task specific. It is relatively unsurprising that such scaffolding helps the model
- No justification as to why this method is better and not another alternative. For example asking the tomi question first, where does the <agent> believe object is and then asking the model about the helping questions are simpler scaffolds that the authors fail to test
    - A 1-shot baseline is needed
    - Methods need to be compared based on # of tokens in the prompt
- Other prompting methods, that are generic to reasoning need to be tested: Eg: a 0-shot detailed tree of thought prompt, detailed self-critique prompt etc.
- As a method being contributed, there need to be  a broader set of evaluations across domains.
- Similar to the results on T4D, there is a lack of results with other language models with FaR.

Citations:

[1] Gandhi, K., Fränken, J. P., Gerstenberg, T., & Goodman, N. D.  (2023). Understanding social reasoning in language models with language  models. *arXiv preprint arXiv:2306.15448*.

[2] Moghaddam, Shima Rahimi, and Christopher J. Honey. "Boosting Theory-of-Mind Performance in Large Language Models via Prompting." *arXiv preprint arXiv:2304.11490* (2023).

[3] Trott, Sean, et al. "Do Large Language Models know what humans know?." *Cognitive Science* 47.7 (2023): e13309.

[4] Jones, Cameron Robert, Sean Trott, and Ben Bergen. "EPITOME: Experimental Protocol Inventory for Theory Of Mind Evaluation." *First Workshop on Theory of Mind in Communicating Agents*. 2023.

[5] Ho, Mark K., Rebecca Saxe, and Fiery Cushman. "Planning with theory of mind." *Trends in Cognitive Sciences* 26.11 (2022): 959-971.

[6] Meta Fundamental AI Research Diplomacy Team (FAIR)†, et al. "Human-level
 play in the game of Diplomacy by combining language models with
strategic reasoning." *Science* 378.6624 (2022): 1067-1074.

[7] Gandhi, Kanishk, Dorsa Sadigh, and Noah D. Goodman. "Strategic Reasoning with Language Models." *arXiv preprint arXiv:2305.19165* (2023).

[8] Dasgupta, Ishita, et al. "Collaborating with language models for embodied reasoning." *arXiv preprint arXiv:2302.00763* (2023).

[9] Hu, Jennifer, and Roger Levy. "Prompt-based methods may underestimate large language models' linguistic generalizations." *arXiv preprint arXiv:2305.13264* (2023).

**Questions:**

Please see the suggestions specified in the weaknesses.

---

> ### Author Response · Authors · 2023-11-19
> **Author Response**
>
> Thank you so much for your helpful feedback and suggestions. We believe all of your concerns can be addressed within the response period. Our responses to weaknesses are as follows:
>
> ### T4D Weakness:
> __[Clearer formulation of the multi-step inference process]__
>
> Thanks for the suggestion. Indeed, for models to succeed at T4D, they need to reason about utilities, ToM inferences, observations, etc. In Section 4.2 (especially Table 2), we break down the multi-step inference process from collected human-written rationales and defined three reasoning challenges: **question decomposition** that captures reasoning about utilities (who benefits from information the most is who needs it most to fulfill a goal), **ToM inferences** that capture reasoning about mental states (A has a false belief of a location of an item), and **common sense assumptions** that capture reasoning about observations (percepts). Moreover, we provide experimental results of the model's performance change when we give hints regarding each challenge. We will articulate this part more clearly.
>
>
> __[T4D which is based on ToMi suffers from the same limitations as ToMi]__
>
> ToMi is a large-scale and well-established benchmark for ToM (also echoed by [2]). We chose to apply T4D on ToMi to better compare with well-grounded related work and for the community to build on. Since we were starting from a new task, building on an existing dataset ensures we only change one variable at a time. Moreover, we acknowledged the limitations of ToMi in our paper by including generalization tests of modified story structure as well as Faux Pas in Section 6.4.
>
> __[Unsurprising that adding “hops“ leads to lower zero-shot performance.]__
>
> Instead of merely adding more hops of reasoning, we devised T4D in a principled way by changing the probing questions from others’ beliefs or actions to the model’s own action decisions based on inferences. Further, Section 4.2 discusses why this shift makes T4D more challenging: the models have to self-discover what to reason about instead of being given hand-holding inference questions as the original ToMi.
>
> __[Alternate probing methods]__
>
> Good suggestion! We tried generative LM probing in the early stage of designing T4D, i.e., asking models to complete a sentence and observing a similar trend. However, we decided on the multiple-choice format because providing options makes the task more well-defined because evaluating free-form generation without probability logit access is error-prone ([1]). Further, adding multiple choices also circumscribes the scope and the goals of the task -- open-ended generation leaves room for fantastical outputs, whereas we are interested in specifically probing the model's capabilities for T4D grounded in the situation.
>
> __[Related work of using theory of mind for planning]__
>
> Thanks for the pointers, we have added them in a new extended related work section in Appendix A due to space constraints. We would like to note that T4D differs from this line of work by exposing a key limitation of LLMs even with short-term social reasoning tasks: connecting inference to action is not trivial; we see this insight as orthogonal to ToM for planning and in a strategic environment.
>
> __[Other theory of mind benchmarks]__
>
> Newer benchmarks such as BigToM are very good studies measuring ToM in LMs. While there is a surface-level similarity in the task setups (both contain “action” probing), there is an important distinction between action probing in BigToM and T4D: probing beliefs of others’ actions such as “*What will A do?*” (BigToM) vs. probing model’s own action decisions based on beliefs of others such as “*Who should you provide help with?*” (T4D).  Specifically, in T4D, models have to treat themselves as situated agents and decide their **own** actions, based on potential reasoning about others’ mental states and predictions of others' actions. We argue that these are complementary directions, and both are important to understand LLMs’ capabilities related to ToM. In summary, T4D builds on previous benchmarks by focusing on models’ capabilities to determine actions for themselves based on mental state reasoning.
>
> (Gentle note that according to the last question in FAQ from ICLR Reviewer Guide (https://iclr.cc/Conferences/2024/ReviewerGuide), “*if a paper was published (i.e., at a peer-reviewed venue) on or after May 28, 2023, authors are not required to compare their own work to that paper.*” and “*they may be excused for not knowing about papers not published in peer-reviewed conference proceedings or journals, which includes papers exclusively available on arXiv.*” and Gandhi et al. 2023. was released on arXiv on June 21, 2023, some other papers in the suggested list are also only on arXiv or released later than May 28, 2023).

---

> ### Author Response · Authors · 2023-11-19
> **Author Response cont.**
>
> __[Probing method restricted to questions about helping others]__
>
> In 6.4, we also include results from T4D-Faux Pas tests, where we convert the original tests from [5] that ask questions of “*Who committed a faux pas?*” to probing action decisions about providing emotional support “*Who should you provide emotional support?*”. In future work, we plan to extend to more diverse domains, including scenarios from BigToM and others.
>
> __[Closed-sourced models]__
>
> We also probed Llama 2-70B on T4D and have included results in the Appendix F in the updated paper. We find a similar trend of models struggling on T4D (22%) significantly more than ToMi (63%).
>
>
>
>
> ### FaR Method Weakness:
> __[Prompt for FaR is much longer and is much more task specific]__
>
> RE length: Prior work has shown that length can both hurt and help LLMs, ([3], [4]) and it is unclear if more complex prompts are always better. Further, we experiment with baselines such as ToT, which have similar length prompts.
>
> RE task-specific: We show that FaR can generalize to other domains different from the original ToMi set in Section 6.4. Furthermore, through analysis in Section 4.2, we find that the key bottleneck of LLMs finding T4D hard is that without hand-holding inference questions such as “what are A’s mental states about X?”, models could not locate the relevant inference to decide an action. Thus, we designed FaR specifically to not leak the mental states and instead of proposing a generalizable reasoning structure about predicting and then reflecting (discussed in Section 5, first 2 paragraphs).
>
> __[Why not alternative such as two-step: first ask mental states, then determine action]__
>
> A key reasoning challenge in T4D is that models must *self-discover* that they should leverage mental state reasoning to decide actions. If asked in a 2-step way, the task will leak about "what to infer about" to LLMs. In contrast, FaR did not leak that models should reason about the mental states and provide a reasoning structure.
>
> __[Other prompting methods like tree-of-thought should be compared]__
>
> Zero-shot tree of thought is one of our baselines, as included in Figure 5, which has a similar number of words as FaR (for both input and output). As for self-critique, our method (FaR) already includes a flavor of self-critique in the Reflect step.
>
> __[Test on other domains]__
>
> We include story structure changes and T4D-Faux Pas generalization tests in 6.4, both of which show consistent improvements from FaR. In our pilot study, we also tested another widely-used ToM test: the smarties test, where the label of a container does not match the content, and found a similar trend in experimental results.​​
>
>
> [1] Shapira, Natalie, et al. "Clever hans or neural theory of mind? stress testing social reasoning in large language models." arXiv preprint arXiv:2305.14763 (2023).
>
> [2] Melanie Sclar, Sachin Kumar, Peter West, Alane Suhr, Yejin Choi, and Yulia Tsvetkov. 2023. Minding Language Models’ (Lack of) Theory of Mind: A Plug-and-Play Multi-Character Belief Tracker. In Proceedings of the 61st Annual Meeting of the Association for Computational Linguistics (Volume 1: Long Papers), pages 13960–13980, Toronto, Canada. Association for Computational Linguistics.
>
> [3] Fu Y, Peng H, Sabharwal A, Clark P, Khot T. Complexity-based prompting for multi-step reasoning. arXiv preprint arXiv:2210.00720. 2022 Oct 3.
>
> [4] Madaan A, Yazdanbakhsh A. Text and patterns: For effective chain of thought, it takes two to tango. Findings of EMNLP, 2023
>
> [5] Shapira, Natalie, Guy Zwirn, and Yoav Goldberg. "How Well Do Large Language Models Perform on Faux Pas Tests?." Findings of the Association for Computational Linguistics: ACL 2023. 2023.

---

> > ### Author Response · Authors · 2023-11-22
> > **Reminder to review response**
> >
> > As the rebuttal period draws to a close, we kindly ask you to review our responses to your comments and queries. We hope that our answers have been satisfactory and that we've addressed all your concerns effectively.
> >
> > Please let us know if there are any additional queries or points that require clarification.

---

> > > ### Author Response · Authors · 2023-11-23
> > > **Friendly reminder to respond to author rebuttal**
> > >
> > > Dear Reviewer WCD3,
> > >
> > > Thank you again for your review! As the rebuttal period draws to a close, we kindly ask you to review our responses to your comments and queries. We hope that our answers have been satisfactory and that we've addressed all your concerns effectively.
> > >
> > > Please let us know if there are any additional queries or points that require clarification. We will be more than happy to reply again!
> > >
> > > Sincerely,
> > >
> > > Paper-8449 Authors

---

> > > > ### Comment · Reviewer_WCD3 · 2023-11-23
> > > > **Response to the authors**
> > > >
> > > > I thank the authors for their comprehensive response. I am glad the added an open-source model; this makes the work more accessible and reproducible. I again summarize my concerns separately for the benchmark and method.
> > > >
> > > > ### Benchmark
> > > > - T4D introduces a different way compared to ToMi to prove models' theory of mind reasoning. It adds a few extra inferences to the ToMi scenarios. I don't buy the argument that "changing the probing questions from others’ beliefs or actions to the model’s own action decisions" does not require added hops and merely changes the probing. When you ask a model to predict an action,  it has to make an added inference to go from the belief to action.
> > > > - Similar to reviewer 6i9F, I don't think that the distinction between self and other's actions is clear. Thank you for pointing me to the guidelines! The newer papers weren't central to my critique.
> > > > - If we assume T4D is a completely new way to probe LMs/ intelligent agents for ToM, just converting ToMi (main focus of the paper) to ToMi - T4D might not be enough. T4D-faux pas, story structure is not discussed till the end of the FaR section. I would love to see T4D being framed as a general way to transform a benchmark (as the authors point out). Eg: Different benchmarks from Shapira et al. [2023], BigToM could be converted to their T4D variants. ToMi by itself is very limiting.
> > > >
> > > > ### Method
> > > >
> > > > My main qualm with the method is that it is not compared to comparable baselines. One look at appendix D shows how FaR has way more structure to the prompt compared to the baselines. A simple analysis of the number of gpt-4 tokens tells us this (text that is bolded in the prompt):
> > > > CoT: 12
> > > > ToT: 60
> > > > Self-Ask: 95
> > > > FaR: 264
> > > >
> > > > Added complexity is okay as long as it is generally applicable.
> > > > So, the added complexity of a method must be justified by a broad set of evaluations. Right now it is limited to ToMi, Faux-Pas that have similar inferences.
> > > >
> > > > I choose to keep my score as the paper could use a round of additional review. I wish the authors the best.
> > > > PS: I apologize for the delay in posting this response.

---

> > > > > ### Author Response · Authors · 2023-11-23
> > > > > **Thanks for replying and quick response**
> > > > >
> > > > > Thank you for your reply. We appreciate you taking the time and are glad that you find our response helpful. As the rebuttal period comes to a close, we would like to make several quick points.
> > > > >
> > > > > __["I don't buy the argument that "changing the probing questions from others' beliefs or actions to the model's own action decisions" does not require added hops and merely changes the probing."]__
> > > > >
> > > > > Indeed, T4D added additional inferences to the ToMi scenarios, but we argue that the added inferences are especially hard due to the questions about models deciding actions for themselves. From the ablations in Section 4.2, we find that not all added inferences lead to such a significant performance drop from ToMi (providing hints on question decomposition and ToM inference makes the task easy). The key bottleneck is that *deciding actions for oneself requires: 1) reasoning about others' mental states and 2) choosing actions that maximize the utilities*. Probes that ask others' beliefs or actions provide a clear target on what inferences should be considered for 1), but for T4D, 1) is left implicit (as mentioned in Section 3.1); thus, models have to figure out themselves what are the salient inference questions. LLMs struggle with this particular added layer of inference, which is the main insight of probing LLMs on T4D.
> > > > >
> > > > > __["A simple analysis of the number of gpt-4 tokens tells us this (text that is bolded in the prompt): CoT: 12 ToT: 60 Self-Ask: 95 FaR: 264"]__
> > > > >
> > > > > We again point to several works that show adding complexity could either help or hurt LLM performance [1, 2] and argue that more token counts do not necessarily imply that methods are not comparable.
> > > > >
> > > > > As for adding structure, this is the main contribution of FaR (discussed in Section 5, paragraph 1): we find injecting proper reasoning structure could largely enhance reasoning for scenarios where models need to decide actions for themselves (echoing the main challenge in T4D). We tested on story generalization and T4D-Faux Pas, both requiring models to choose actions as situated agents, but in different scenarios, and we find the improvement is consistent.
> > > > >
> > > > > [1] Fu Y, Peng H, Sabharwal A, Clark P, Khot T. Complexity-based prompting for multi-step reasoning. arXiv preprint arXiv:2210.00720. 2022 Oct 3.
> > > > >
> > > > > [2] Madaan A, Yazdanbakhsh A. Text and patterns: For effective chain of thought, it takes two to tango. Findings of EMNLP, 2023

---

### Official Review · Reviewer_pfbS · 2023-11-01

**Soundness:** 3 good
**Presentation:** 3 good
**Contribution:** 3 good
**Rating:** 6
**Confidence:** 3

**Summary:**

This paper first proposes a new evaluation paradigm for LLM and also proposes a new prompting framework for LLMs called FaR, which provides a structure to encourage LLMs to anticipate future challenges and reason about potential actions. Compared with other prompting methods such as Chain-of-Thought and Self-Ask, FaR boosts the LLM (e.g., GPT-4) performance significantly. Besides, it also outperforms other baseline methods in few-shot in-context learning, showing the effectiveness of the new prompting method.

**Strengths:**

1. The new prompting framework that encourages LLMs to anticipate future challenges and reason about potential actions is novel and impressive.

2. The paper is well-organized and well-written with clear motivations, detailed discussion, nice figures, and sufficient comparison experiments, making it easy to follow and understand.

3. This work performs comprehensive experiments over benchmark data to show the effectiveness of Far in several in-context settings.

**Weaknesses:**

1. This work illustrates the differences among FaR and GPT-4 in real-world scenarios. I am also curious about what it will look like with different prompting methods, such as Chain-of-Thought and Tree-of-Though. This work does not introduce existing prompting methods in detail, which will confuse the audience to some degree.

2. FaR breaks the T4D process into three steps, question decomposition, theory-of-mind inference, and common sense assumption. However, the motivations for the breakdown have not been introduced in this work. I would like to suggest that this work explains the motivations clearly.

**Questions:**

1. What are the differences between existing prompting methods and FaR in LLMS?  What are the advantages of FaR?
2. What is the motivation for the three steps in the T4D process?

---

> ### Author Response · Authors · 2023-11-18
> **Author Response**
>
> Thank you so much for your helpful feedback and suggestions. Our responses to weaknesses and questions are as follows:
>
> __[Curious about what it will look like with different prompting methods, such as Chain-of-Thought and Tree-of-Though]__
>
> We did include comparisons with chain-of-thought and tree-of-thought prompting methods as shown in Figure 5 and explained in detail in Section 6.1. We also included the full prompts in Appendix D.
>
>
> __[FaR breaks the T4D process into three steps. However, the motivations for the breakdown have not been introduced in this work.]__
>
> The three steps are our attempts to analyze what makes T4D challenging. We arrived at the three reasoning challenges from collecting human-written rationales as explained in detail in Section 4.2 and Appendix C. Additionally, through experiments, we found that providing hints regarding each challenge results in very different LLM performance, which helped us speculate that the key reason why T4D is hard for LLMs is that models struggle to locate relevant inference evidence to determine their actions. FaR actually did not incorporate the 3 steps as it would leak the reasoning process. Instead, FaR proposes a generalizable reasoning structure with two steps: Foresee and Reflect that helps models to arrive at relevant inferences.

---

> > ### Author Response · Authors · 2023-11-22
> > **Reminder to review response**
> >
> > As the rebuttal period draws to a close, we kindly ask you to review our responses to your comments and queries. We hope that our answers have been satisfactory and that we've addressed all your concerns effectively.
> >
> > Please let us know if there are any additional queries or points that require clarification.

---

> > > ### Author Response · Authors · 2023-11-23
> > > **Friendly reminder to respond to author rebuttal**
> > >
> > > Dear Reviewer pfbS,
> > >
> > > Thank you again for your review! As the rebuttal period draws to a close, we kindly ask you to review our responses to your comments and queries. We hope that our answers have been satisfactory and that we've addressed all your concerns effectively.
> > >
> > > Please let us know if there are any additional queries or points that require clarification. We will be more than happy to reply again!
> > >
> > > Sincerely,
> > >
> > > Paper-8449 Authors

---

> > > > ### Comment · Reviewer_pfbS · 2023-11-23
> > > > **Thank you for your clarification!**
> > > >
> > > > Thank you for your clarification and I will insist on supporting this work.

---

### Author Response · Authors · 2023-11-19
**Thanks for your feedback!**

We thank the reviewers for their thoughtful comments. We are delighted that they found our work well-motivated, insightful, and comprehensive. We believe that the main concerns of the reviewers can be addressed during the response period. We have attempted to address reviewers' concerns and look forward to a fruitful discussion.

We also updated the paper pdf to reflect on reviewers' suggestions with main changes in Appendix A (extended related work) and F (additional experiments) highlighted in blue text.

---

### Meta-Review · Area_Chair_PPKk · 2023-12-13

**Metareview:**

The paper proposes a new benchmark, Thinking for Doing (T4D), to test the theory of mind capabilities in large language models (LLMs). It modifies the existing ToMi benchmark to require models to not just infer mental states but act based on those inferences. Experiments are conducted with GPT-4 and PaLM-2.

The reviewers' main concerns are that the empirical part is not convincing. Although the authors have tried their best to address reviewers' concerns. However, the reviewer still thinks that this paper needs significant modification before acceptance.

**Justification For Why Not Higher Score:**

the empirical part is not convincing

**Justification For Why Not Lower Score:**

n/a

---

### Decision · Program_Chairs · 2024-01-16

Reject